# Spatial-Temporal Evolution and Influencing Mechanism of Traffic Dominance in Qinghai-Tibet Plateau

Dongchuan Wang, Kangjian Wang, Zhiheng Wang *, Hongkui Fan, Hua Chai, Hongyi Wang, Hui Long, Jianshe Gao and Jiacheng Xu

School of Geology and Geomatics, Tianjin Chengjian University, Tianjin 300384, China
* Correspondence: wangzhiheng@tcu.edu.cn

**Abstract:** Transportation is an important resource for the sustainable development of the Qinghai-Tibet Plateau. It is of great practical significance to evaluate and study the law and mechanism of spatial and temporal differentiation of traffic dominance degree. Based on the methods of the Origin-Destination cost matrix, least squares method, and geographically weighted regression, this paper establishes a traffic dominance evaluation system at the county scale and discusses the spatial pattern and influence of traffic dominance in the Qinghai-Tibet Plateau from 2015 to 2019. The results show that: (1) The overall traffic construction of the Qinghai-Tibet Plateau has been accelerated, and the traffic accessibility between counties has been significantly enhanced; (2) The traffic dominance of the Qinghai-Tibet Plateau is significantly different from east to west, and the central area, with "Xining-Lhasa" as the axis, expands to the outer circle with an irregularly decreasing spatial pattern; and (3) The effect of rapid urbanization development and population carrying capacity enhancement on the traffic dominance of the Qinghai-Tibet Plateau has gradually increased, and the effect of elevation has been weakening from 2015 to 2019.

**Keywords:** traffic dominance; spatial pattern; geographically weighted regression model; influence mechanism; Qinghai-Tibet Plateau

## 1. Introduction

As the premise and important support of regional social and economic development, transportation infrastructure reshapes the regional spatial structure by changing location conditions [1,2]. Traffic dominance is the basic element of transportation infrastructure that plays a role in regional economic and social development. Studies on the development, accessibility, and superiority of regional transportation networks have always been the focus of transportation geography, urban and rural planning, and other disciplines. The Qinghai-Tibet Plateau is the roof of the world, the water tower of Asia, and the third pole of the earth and will play an important strategic role in China's ecological environment [3,4], economic development, and opening in terms of geographical location, natural conditions, and economic resource advantages [5,6]. The construction and improvement of the transportation infrastructure on the Qinghai-Tibet Plateau can change the local location conditions, and thus have a profound impact on the regional economic development, industrial layout, and spatial structure, as well as promote the coordinated development of human-land relations [7–9]. The impact of the development of transportation infrastructure on the Qinghai-Tibet Plateau is multi-faceted. While driving the GDP growth rate of the counties along the line and promoting the development of local tourism and culture and other related industries, it also has a negative impact on the environment [10]; for example, increasing the vulnerability of the ecological environment [11], affecting the stability of landscape patterns [12], and disturbing the living rules and habits of wild animals [13]. These are all problems that cannot be ignored in the process of transportation infrastructure

construction. Therefore, it is of great importance to study the development of characteristics, laws, and influential factors of the transportation network in the Qinghai-Tibet Plateau from the perspective of human-land coordination.

Traffic dominance is an integrated index to measure the level of regional traffic superiority and is a comprehensive reflection of the supplementary capacity of transportation, urban proximity, regional external connection ability, and regional accessibility [14,15]. Compared with single indicators, such as transportation accessibility road density, etc., it can better reflect the level of traffic development for a region overall. The measurement of traffic dominance and the spatio-temporal analysis for traffic dominance can effectively evaluate the regional infrastructure construction and future development potential.

The concept of traffic dominance was first proposed by Jin [16]. Subsequently, many scholars have carried out significant research on the construction, integration method, and spatial distribution mode of traffic dominance. According to the specific situation in Northeast China, Sun et al. [17] introduced transportation facilities and external development factors to explore the spatial differentiation of traffic dominance. Kowk et al. [18] formulated sustainable transportation development indicators. Wang et al. [19] evaluated the differences in traffic dominance in different regions of Shandong Province based on the principle of the traffic superiority index system. Meanwhile, some scholars have also researched the coordinated development of the transportation system and socioeconomic development. [20–22]. Scholars have also carried out fruitful research on the spatial effects of traffic dominance, such as socio-economic development [23], urbanization [24], land efficiency [25], and land suitability [26]. Cheng, Jia, and Sun [27–29] believe that transportation accessibility is positively correlated with industrial agglomeration, and the development of transportation can bring significant economic benefits; Zhu et al. [30] believe that the imbalance of transportation development will restrict the development of urbanization; Cui et al. [31] have studied the spatial relationship between traffic dominance degree and land use efficiency in the Shandong Peninsula. In general, most studies focus on accessibility evolution [32,33], spatio-temporal collapse, and spatial convergence effects brought about by transportation network improvement at the regional scale, and mainly focus on areas with rapid transportation infrastructure construction, a high level of urbanization, and social and economic development, such as the Beijing-Tianjin-Hebei Urban agglomeration [34–36], Greater Bay Area [37–39], Changsha-Zhuzhou-Xiangtan Urban agglomeration [40], Chengdu-Chongqing urban agglomeration, etc. [41]. Restricted by factors such as a fragile ecology and complex natural environment, the development of transportation on the Qinghai-Tibet Plateau has been in a backward state for a long time and has become one of the important factors restricting social and economic development and strengthening external relations [10]. At the same time, the development characteristics and laws of the regional transportation network are an important entry point for the coordinated development of the human-land relationship, and it is of great significance to research the transportation network in the Qinghai-Tibet Plateau [9]. At present, there is still a lack of systematic research on the temporal and spatial evolution mechanism of transportation advantages in ecologically fragile areas such as the Qinghai-Tibet Plateau.

In this paper, road density, the influence of trunk lines, and spatial accessibility are combined to construct an evaluation system of traffic dominance degree. Additionally, the spatial pattern of the evolution characteristics of the traffic dominance degree in the Qinghai-Tibet Plateau are quantitatively analyzed using the data from the road network, railway network, and airport in 2015 and 2019. Then, considering the influence of social and economic factors, the least squares method and a geographically weighted regression model are used to study the influencing factors and the mechanism of traffic dominance in the Qinghai-Tibet Plateau. The study results can provide important scientific references for the optimization of the transportation network layout and high-quality sustainable development in the Qinghai-Tibet Plateau.

## 2. Data Sources and Research Methods

### 2.1. Study Area

The Qinghai-Tibet Plateau is the highest in the world, known as the "roof of the world". It starts from the Himalayas in the south, the Kunlun Mountains and Qilian Mountains in the north, the Pamir Plateau and Karakoram Mountains in the west, and the Qinling Mountains and Loess Plateau in the east (Figure 1). The Qinghai-Tibet Plateau is about 2700 km wide from east to west and 1400 km long from north to south, with a total area of about 2.5 million square kilometers [42,43]. Its unique topography brings great difficulties to the construction of transportation infrastructure.

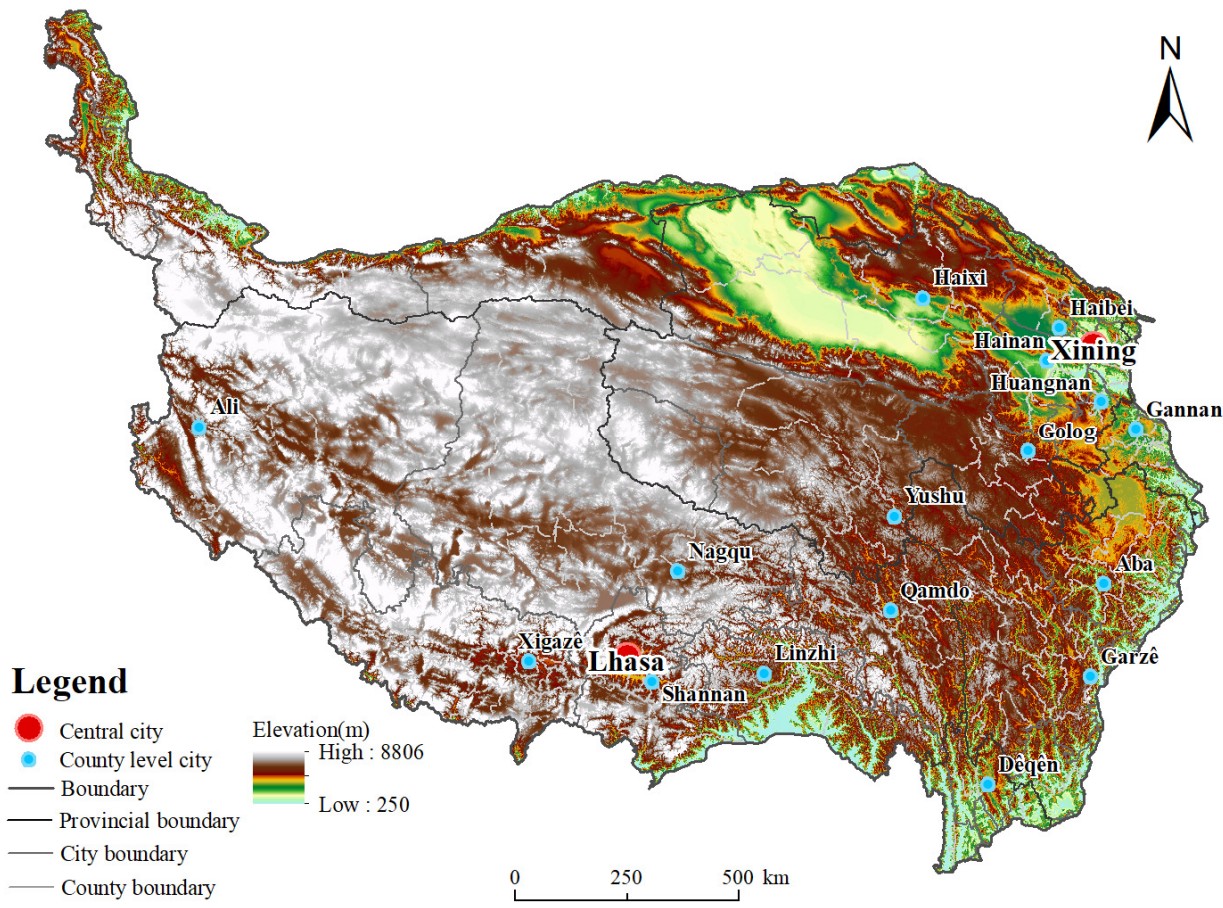

**Figure 1.** Overview of the study area.

Transportation in the Qinghai-Tibet Plateau has lagging infrastructure development, which is characterized by low road network density, single transportation mode, and low network quality [44]. Since the implementation of the Belt and Road policy, the traffic network construction of the Qinghai-Tibet Plateau has been expanding. In 2015, nine key highway projects such as Yatong and Chage were completed and opened, and the construction of "rural roads with four good" has been strengthened. The total mileage of rural roads in the province has reached 615,000 km, effectively improving the patency rate between townships. At the same time, the successive construction of several high-grade roads such as the G6 highway (Golmud-Lhasa Section) and the Gongyu highway has greatly strengthened the links between the Qinghai-Tibet plateau and the mainland. However, the remote northwest areas are mostly affected by topography and terrain, and development of the transportation is relatively slow. Equilibrating the pace of traffic construction between various regions of the Qinghai-Tibet Plateau is one of the key directions of future research on traffic dominance.

## 2.2. Data Sources

The data used in this research include (1) two phases of traffic road data in the Qinghai-Tibet Plateau in 2015 and 2019 (Figure 2); (2) the basic geographic data in the study area from mainly elevation data; (3) population density data in Qinghai-Tibet Plateau in 2015 and 2019 (Figure 3); and (4) social and economic development data such as urban construction area and public financial expenditure in 2015 and 2019. The specific data sources are shown in Table 1.

This paper is based on the national OSM road vector data in 2015 and 2019 which includes four levels: the level of highway, national highway, provincial, and county highway.

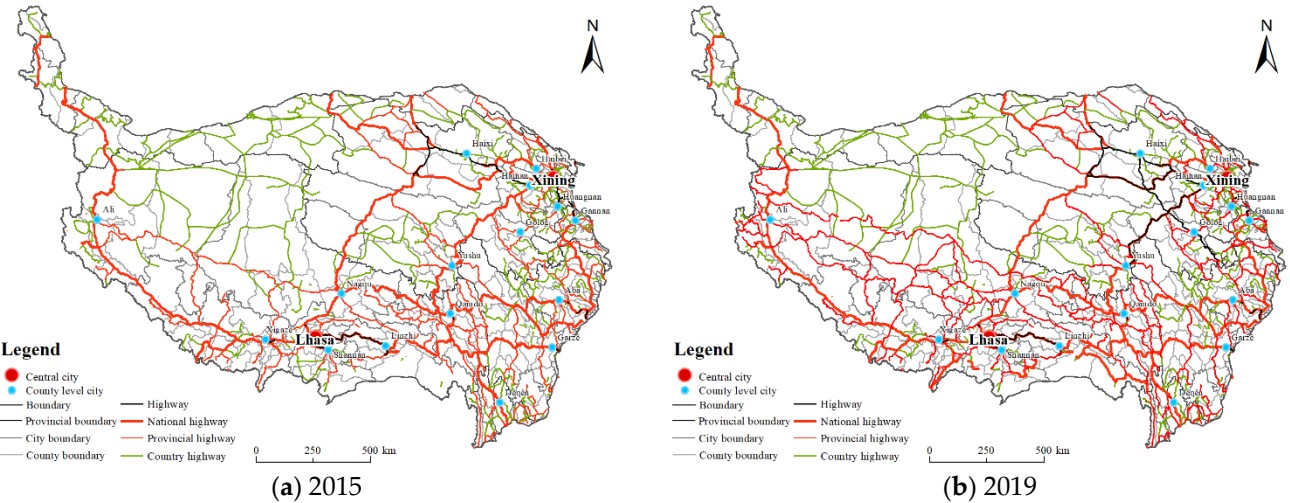

**Figure 2.** Traffic Road Network in Qinghai-Tibet Plateau in 2015 and 2019. (**a**) distribution map of road network in 2015; (**b**) distribution map of road network in 2019.

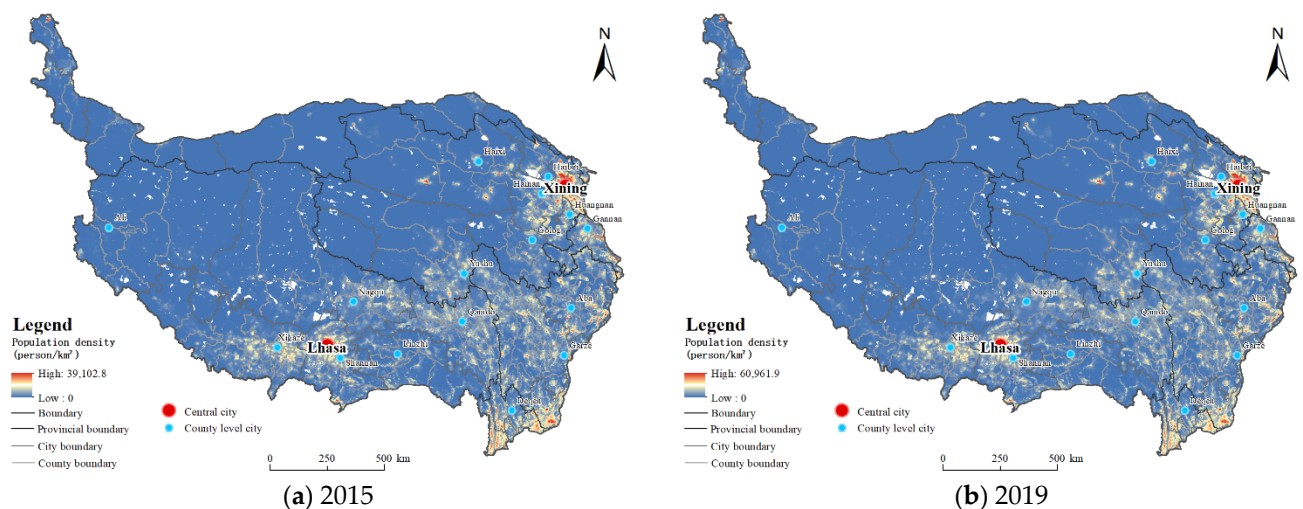

**Figure 3.** Population density of Qinghai-Tibet Plateau in 2015 and 2019. (**a**) distribution map of population density in 2015; (**b**) distribution map of population density in 2019.

| Data Type | Data Sources | Explanation |
|---|---|---|
| National highway/Provincial/Country highway/Highway | Open Street Map (OSM) database | - |
| Passenger station/Railway station /Highway intersection/Airport | Baidu Map POI | - |
| Elevation | Geospatial Data Cloud http://www.gscloud.cn/ (accessed on 18 June 2020) | DEM extraction from Qinghai-Tibet Plateau based on SRTM DEM 30 m resolution digital elevation data. |
| Population density | World Pop https://www.worldpop.org/ (accessed on 11 May 2022) | Based on the downloaded 1 km population density grid data of the Qinghai-Tibet Plateau in 2015 and 2019; the population density data at the county unit scale are obtained by zoning statistics. |
| Socio-economic indicators | *China Statistical Yearbook (County-Level) (2016, 2020)* [1] *China City Statistical Yearbook (2016, 2020)* [2] | Urban construction area, public fiscal expenditure, fixed asset investment, regional GDP, and employees of secondary and tertiary industries in each county unit. |

[1] *China Statistical Yearbook (County-Level)* (2016, 2020) includes the basic information, comprehensive economy, agriculture, industry, education, health, social security, etc. of more than 2000 county-level units in China in 2015 and 2019. [2] *China City Statistical Yearbook* is an annual statistical publication. *China City Statistical Yearbook* (2016, 2020) includes the main statistical data on the socio-economic development of cities at all levels across the country in 2015 and 2019.

## 2.3. Research Methods

Based on the evaluation of traffic advantages in recent years and related research ideas [16,45,46], this paper utilized the measurement model of transportation dominance to quantitatively analyze the pros and cons of traffic at the county level in time and space and took advantage of the geographically weighted regression model based on the least squares regression model to explore the influencing factors and spatial heterogeneity of the overall traffic dominance of the Qinghai-Tibet Plateau.

### 2.3.1. Road Density

Road density refers to the ratio of traffic line length to county area. Studies have found that the greater the road density, the stronger the regional economic development and external ties, and the stronger the ability to protect the regional economy [17,47]. The greater the density of the county road network, the better the traffic conditions of the county. Its expression is:

$$D_i = \sum_{j=1}^{m} P_{ij} \frac{L_{ij}}{Area_i}, i \in (1, 2, 3, \ldots, n) \tag{1}$$

where $D_i$ is the road density, $m$ is the type of transport, $P_{ij}$ is the proportion of the *j*-th transportation mode in the *i*-th county, $L_{ij}$ is the length of the *j*-th transportation mode in the *i*-th county, and $Area_i$ is the area of the *i*-th county. According to the actual situation of the Qinghai-Tibet Plateau, the highway, national highway, provincial, and county highway are assigned 2.0, 1.5, 1.0, and 0.5, respectively.

### 2.3.2. Influence Degree of Trunk Lines

The influence degree of trunk lines refers to the level of transportation facilities in the research area and the degree of influence on the development of the area. The greater the influence of the trunk line, the stronger the support ability of the development system for the area. Existing studies [48] usually take roads and railways as empowerment types and do not fully consider the increasing proportion of aircraft travel in recent years. Based on the research results of Wang, Guo, etc. [49,50], and considering the characteristics of the Qinghai-Tibet Plateau, this paper adopts the method of assigning and scoring to evaluate the traffic facilities (Table 2).

**Table 2.** Weight assignment of traffic facilities.

| Type | | Standard | Weight Assignment |
|---|---|---|---|
| Road | Highway | Contains highway exits | 2.5 |
| | | Lh* ≤ 30 km | 2.0 |
| | | 30 km < Lh ≤ 60 km | 1.0 |
| | | Lh > 60 km | 0 |
| | Ordinary highway | 1st | 2.5 |
| | | 2nd | 2.0 |
| | | 3rd | 1.5 |
| | | Others | 1.0 |
| | Railway | Railway station | 2.5 |
| | | Lh ≤ 30 km | 2.0 |
| | | 30 km < Lh ≤ 60 km | 1.0 |
| | | Lh > 60 km | 0 |
| | Airport | Airport | 2.0 |
| | | Lh ≤ 30 km | 1.0 |
| | | 30 km ≤ Lh ≤ 60 km | 0.5 |
| | | Lh > 60 km | 0 |

* Lh refers to buffer distance from the transportation infrastructure.

### 2.3.3. Spatial Accessibility

Spatial accessibility represents the difficulty of residents in an area to reach their destination and can reflect the accessibility between two places. The accessibility consideration takes the transportation distance between nodes and the influence of potential factors as important measurement indicators. This paper refers to previous research [51], and adopts the shortest distance model to determine the accessibility of each county-level administrative division. Its expression is:

$$A_i = \sum_{j=1}^{n} I_{ij} \ i, j \in (1, 2, 3, \ldots, n) \tag{2}$$

where $A_i$ is the reachability coefficient of node $i$, $I_{ij}$ is the shortest distance between two nodes, and $n$ is the total number of nodes in the region. The smaller the value of the regional accessibility coefficient $A_i$ is, the better the accessibility is.

The accessibility of the whole traffic network can be defined by Equation (2). The expression is:

$$A = \sum_{i=1}^{n} A_i \ i \in (1, 2, 3, \ldots, n) \tag{3}$$

where $A$ is the accessibility coefficient of traffic space in the whole region, $n$ is the total number of nodes in the region, $i$ is the $i$-th node, and $A_i$ is the accessibility coefficient of node $i$.

The regional reachability coefficient is the ratio of the reachability measure of each node to the average value of the reachability measure of all nodes in the region. The expression is:

$$R_i = \frac{A_i}{(A/n)} \ i \in (1, 2, 3, \ldots, n) \tag{4}$$

where $R_i$ is the accessibility coefficient, which can express the relative accessibility level of each node in the whole network. The smaller the accessibility coefficient is, the higher the accessibility is, and vice versa.

### 2.3.4. Ordinary Least Squares (OLS)

OLS regression analysis can effectively explore the contribution of traffic factors to traffic [52]. In view of the special spatial difference of traffic dominance in the Qinghai-Tibet Plateau, it is not affected by a single factor. This paper selects the ordinary least squares

method to analyze the relationship between traffic dominance and multiple factors. The expression is as follows:

$$y = x\beta + \varepsilon \tag{5}$$

where $y$ is the dependent variable, $x$ is the independent variable, $\beta$ is the model coefficient estimated by the data, and $\varepsilon$ is the model residual and obeys the $N(0, \sigma^2)$ distribution.

$\beta$ can be estimated by the method where the sum of squares of deviations between the dependent variable and the predicted value is the smallest, and its expression is:

$$\beta = \left(X^T X\right)^{-1} X^T Y \tag{6}$$

where $\beta$ is the estimated value of parameter $\beta$, $X$ is the independent variable matrix, and $Y$ is the dependent variable matrix.

2.3.5. Geographically Weighted Regression (GWR)

The geographically weighted regression model is an effective model to deal with the spatial non-stationary phenomenon in regression analysis [53]. It can detect the extent to which the influence factors and the interaction between factors can explain the spatial differentiation of traffic dominance and the spatial matching degree between the influence factors and the change of traffic dominance [54]. When the regression coefficient in the spatial regression model does not change with spatial location, the spatial regression model is a global model. However, due to spatial heterogeneity and spatial non-stationarity, the relationship between independent variables and dependent variables in different spatial subregions is likely to be different. Therefore, it is necessary to deal with the local spatial regression method of spatial heterogeneity, whose expression is:

$$y_i = \beta_0(u_i, v_i) + \sum_{k=1}^{P} \beta_k(u_i, v_i) x_{ik} + \varepsilon_i \tag{7}$$

where $y_i$ is the traffic dominance index, $\beta_0$ is a constant, $(u_i, v_i)$ is the geographic coordinates of the $i$ sample, $\beta_k(u_i, v_i)$ is the $k$-th regression parameter on the $i$ sample, $x_{ik}$ is the $k$-th parameter on the I sample, and $\varepsilon_i$ is random error.

## 3. Results

*3.1. Spatial and Temporal Evolution Characteristics of Traffic Dominance*

3.1.1. Road Density

The high-density areas of the traffic network in the Qinghai-Tibet Plateau are mainly concentrated in the Xining urban agglomeration and Lhasa central agglomeration, and the surrounding areas show a decreasing trend (Figure 4). Affected by the rapid construction of highways, road density in central and northern Qinghai Province increased significantly from 2015 to 2019. With the successive construction of several provincial roads, such as S520 (Banying Section), S302 (Dingshi Section), and S208 (Qiangya Section), the road network density in the southwestern Qinghai-Tibet Plateau has increased.

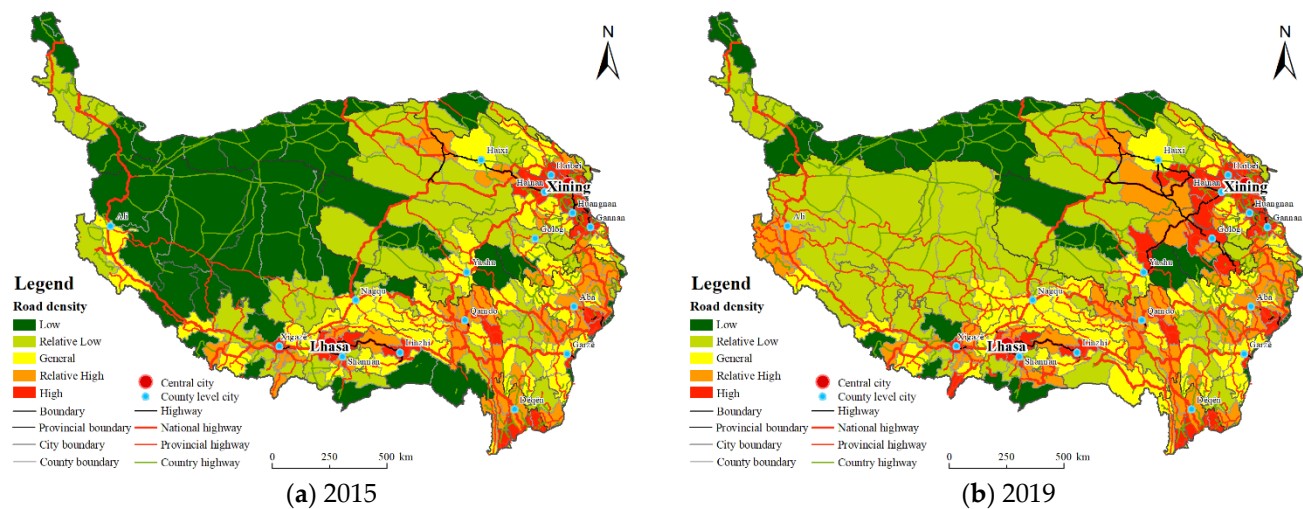

**Figure 4.** Road density in Qinghai-Tibet Plateau in 2015 and 2019. (**a**) distribution map of road density in 2015; (**b**) distribution map of road density in 2019.

### 3.1.2. Influence Degree of Trunk Lines

The influence degree of trunk lines between counties in the Qinghai-Tibet Plateau is significantly different and unevenly distributed, showing a spatial pattern of decreasing radiation from the central city to the outside (Figure 5). The areas with the high influence degree of the main line are the urban agglomeration areas such as Xining and Lhasa, and the surrounding counties are mostly areas with a relatively high influence degree. Areas with relatively weak and weak impact degrees are concentrated in the central and western Qinghai-Tibet Plateau. From 2015–2019, railway stations, such as Yuka Station and Muli Station, and high-grade highways, such as Dema highway and Gongyu highway, were successively constructed, and the influence degree of trunk lines in the central and southern Qinghai-Tibet Plateau showed a significant increase.

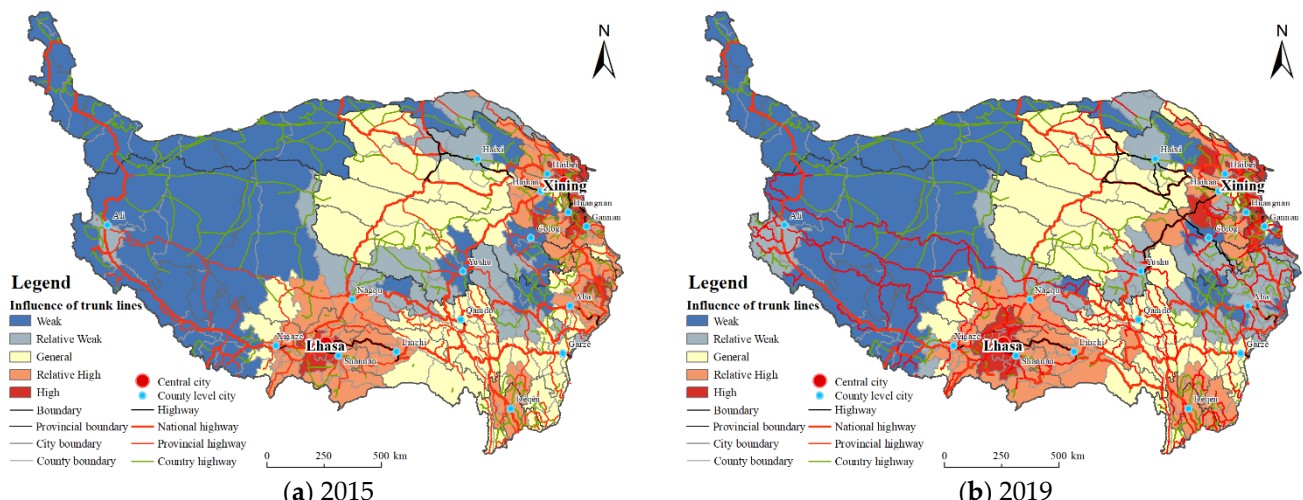

**Figure 5.** Influence degree of trunk lines in the Qinghai-Tibet Plateau in 2015 and 2019. (**a**) distribution map of the influence degree of trunk lines in 2015; (**b**) distribution map of the influence degree of trunk lines in 2019.

### 3.1.3. Spatial Accessibility

Affected by traffic roads and urban distribution, the areas with better accessibility to the Qinghai-Tibet Plateau are concentrated in the core circle with the Xining-Lasa axis (Figure 6). From 2015 to 2019, due to the vigorous construction of national and provincial roads, the spatial accessibility of counties Shuanghu, Luozha, Longzi, and Chayu increased

significantly; however, the western and northern regions of the Qinghai-Tibet Plateau, including Hotan, Kashgar, and other regions, are affected by terrain, climate, etc., with a high cost of transportation, so accessibility is difficult to improve.

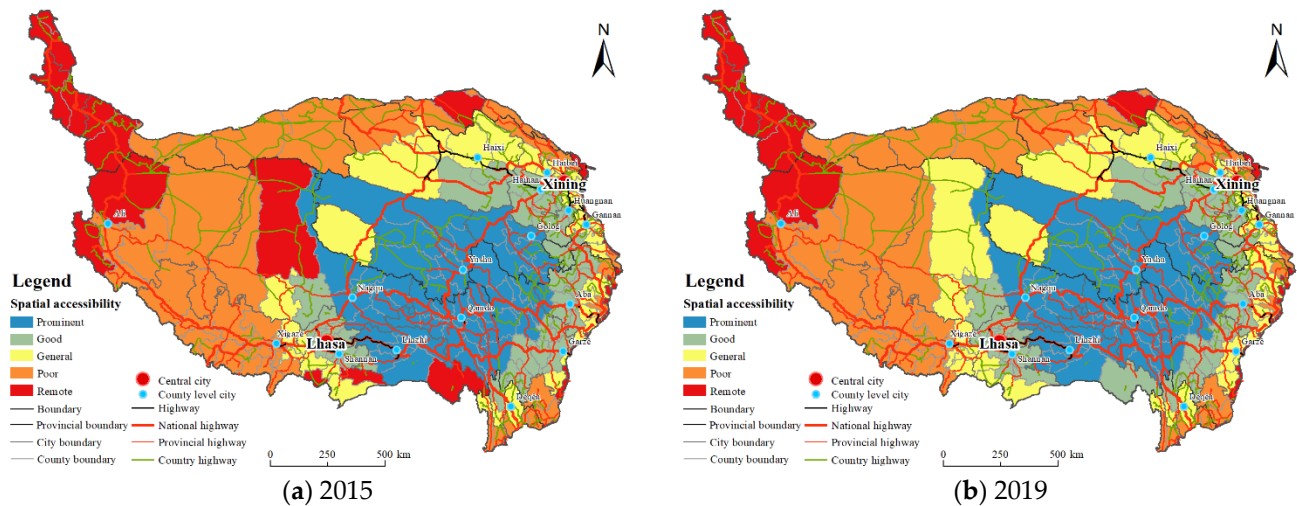

**Figure 6.** Spatial accessibility of Qinghai-Tibet Plateau in 2015 and 2019. (**a**) distribution map of spatial accessibility in 2015; (**b**) distribution map of spatial accessibility in 2019.

### 3.1.4. Traffic Dominance

Based on the indicators of road density, influence degree of trunk lines, and spatial accessibility, this paper uses the analytic hierarchy process (AHP) to assign the weight of each indicator, which is 0.088, 0.243, and 0.669, respectively, and obtains the traffic dominance of county unit by weighted superposition. Overall, the spatial pattern of traffic dominance degree at the county scale in the Qinghai-Tibet Plateau shows the spatial characteristics of the "circle structure" of the "axis-radial". As shown in Figure 7, compared with 2015, the number of counties with low levels of traffic dominance decreased from 49 to 45 in 2019, the number of counties with relatively high levels increased from 40 to 44, and the number of counties with a high level increased from 39 to 44, indicating that the effects of transportation infrastructure construction on the Qinghai-Tibet Plateau in the two periods were remarkable.

From 2015 to 2019, the successive construction of high-grade highways, such as the G219 (Tibet section) and the S208 (Qiangya section), and the increase in the throughput of transportation hubs, such as airports and railway stations in the central urban circle led by Lhasa, the traffic advantage in the central and southern regions of the Qinghai-Tibet Plateau in 2019 was significantly higher than that in 2015. As shown in Figure 8, traffic development and construction in remote areas such as Ali, Akto, and Yecheng are subject to multiple constraints, resulting in a low level of traffic dominance for a long time. The main reason is that the traffic dominance of the Qinghai-Tibet Plateau is not only related to traffic infrastructure construction but is also affected by the social economy, urban distribution and population density, land development, and other factors.

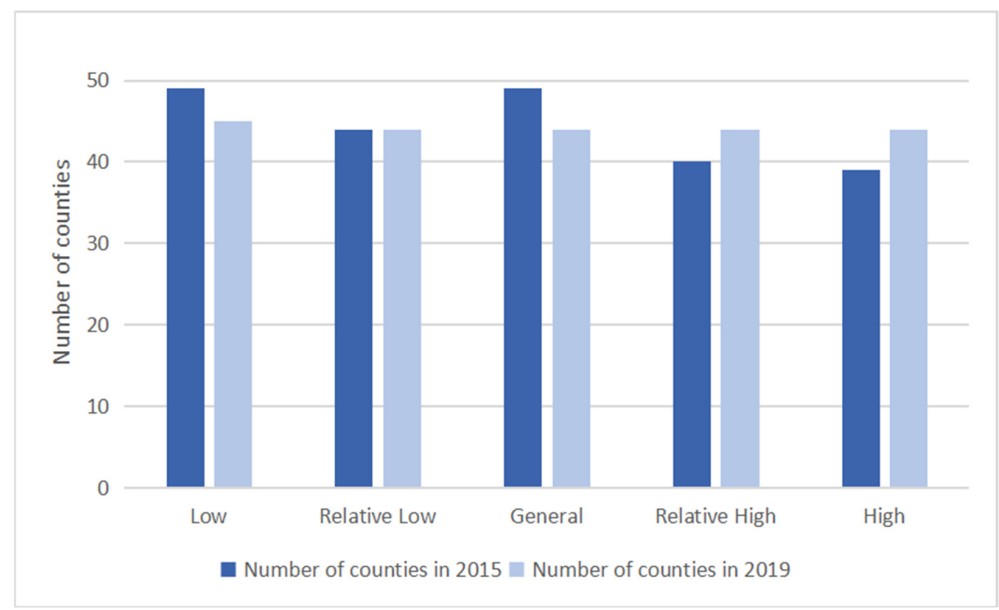

**Figure 7.** Statistics of traffic dominance at the county level in 2015 and 2019.

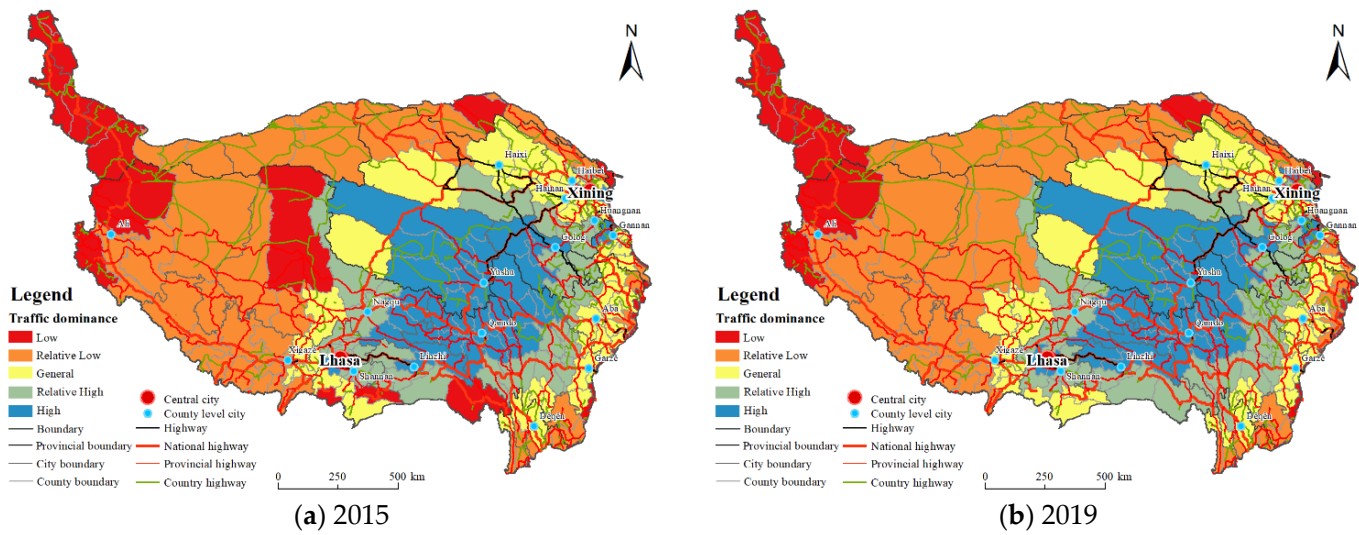

(**a**) 2015 (**b**) 2019

**Figure 8.** Traffic dominance of Qinghai-Tibet Plateau in 2015 and 2019. (**a**) distribution map of traffic dominance in 2015; (**b**) distribution map of traffic dominance in 2019.

### 3.2. Influencing Factors of the Spatial and Temporal Pattern Differentiation of Traffic Dominance

3.2.1. Analysis of Influencing Factors

Traffic dominance is the result of the interaction between the natural environment, land resources, and social development. Considering the previous research [17,21,32,33,41], this paper selects seven factors, which are urban construction area, public financial expenditure, fixed investment, population density, per capita GDP, second and third industry employees, and elevation (Figure 9), and further reveals the contribution of each factor to the traffic dominance of Qinghai-Tibet Plateau and its influencing mechanism.

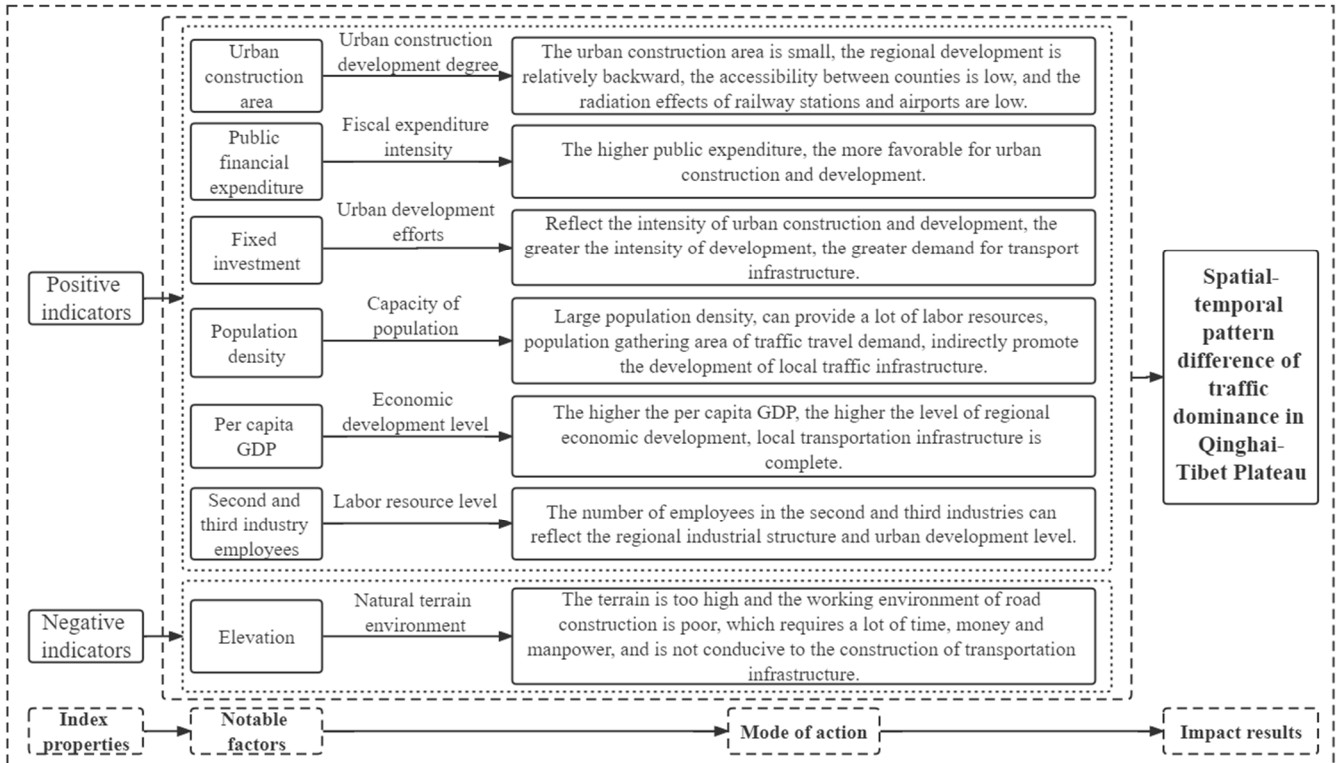

**Figure 9.** Influence factors on the spatial differentiation of traffic dominance.

Negative factors, such as elevation, turn into positive factors when standardized. The OLS regression results show that the VIF values are lower than 10, which meets the requirements of regression analysis. $R^2$ in 2019 can better explain the explained variables, while the residual error of fitting results in 2015 is large. Then, the variables with serious collinearity are eliminated. Finally, elevation, urban construction area, and population density are selected as three factors that have a linear relationship with traffic dominance. The OLS regression results are shown in Table 3.

**Table 3.** The OLS regression analysis results of influencing factors.

| Years | Dimension | Index | Regression Coefficients | Standard Deviation | t-Statistic | Significance ρ |
|-------|-----------|-------|-------------------------|--------------------|-------------|----------------|
|       | Natural Environment | Elevation | −0.317 | 0.066 | −4.817 | 0.00001 |
| 2015  | Land Resources | Urban Construction Area | 0.254 | 0.083 | 3.062 | 0.00250 |
|       | Social Development | Population Density | 0.662 | 0.163 | 4.059 | 0.00000 |
|       | Natural Environment | Elevation | −0.305 | 0.063 | −4.834 | 0.00002 |
| 2019  | Land Resources | Urban Construction Area | 0.335 | 0.076 | 4.377 | 0.00003 |
|       | Social Development | Population Density | 0.716 | 0.144 | 4.968 | 0.00000 |

To further explore the influence of relevant indicators on the traffic dominance of the Qinghai-Tibet Plateau, GWR is used to estimate the parameters of the three dominant factors of elevation, urban construction area, and population density in 2015 and 2019. The regression results are shown in Table 4 and Figure 10. The results show that the adjusted $R^2$ in 2015 and 2019 is 0.414 and 0.556, respectively. The explanatory power of the three indicators on the traffic dominance of the Qinghai-Tibet Plateau has increased year by year, and the influence effect shows a significant indigenous trend. Among them, the regression coefficient of the urban construction area and population density increased significantly in the two periods, indicating that with the rapid development of urbanization, both have a positive role in promoting the traffic dominance of the Qinghai-Tibet Plateau. Overall,

the influence of the three factors on the traffic dominance of the Qinghai-Tibet Plateau has been changing in time and space, and there are different spatial distribution patterns.

**Table 4.** The GWR of traffic dominance and influencing factors in Qinghai-Tibet Plateau.

| Years | Dimension | Regression Coefficients | Adjusted $R^2$ |
|---|---|---|---|
| | Elevation | −0.441 | |
| **2015** | Urban Construction Area | 0.203 | 0.414 |
| | Population Density | 0.677 | |
| | Elevation | −0.421 | |
| **2019** | Urban Construction Area | 0.241 | 0.556 |
| | Population Density | 0.790 | |

From the elevation of the natural environment dimension (Figure 10a,b), the negative impact of elevation on traffic dominance in the Qinghai-Tibet Plateau is weakening year by year. From 2015 to 2019, the regression coefficient was highly aggregated from the central and western regions and decreased to both sides; the spatial distribution pattern of traffic dominance was significantly different as well. The number of highly negatively correlated counties in 2019 was significantly reduced compared with 2015. The difference in the elevation regression coefficient between the two periods decreased, indicating that the influence of elevation on traffic dominance is weakening. This is due to the rapid construction of airports and passenger stations in the Qinghai-Tibet Plateau and the remarkable improvement of land traffic conditions since the proposal of the Belt and Road Initiative. As an important influencing factor to traffic dominance, elevation has constraints on the traffic infrastructure construction and traffic development in the Qinghai-Tibet Plateau.

From the perspective of urban construction areas in the dimension of land resources (Figure 10c,d), the traffic development in the Qinghai-Tibet Plateau is still dominated by the developed eastern urban construction areas. From 2015 to 2019, the overall regression coefficient showed a decreasing trend from east to west in space, and the high correlation areas changed to the southeast aggregation trend in 2019, indicating that the traffic dominance of the Qinghai-Tibet Plateau has a high sensitivity to the urban construction area. From the time dimension perspective, urbanization development and traffic network construction in the Qinghai-Tibet Plateau have developed rapidly from 2015 to 2019 and is the rapid rise period of the urban construction area which mainly decreases from the built-up area with a higher urbanization level in the eastern region to the areas with lower urbanization levels. With the improvement of transportation infrastructure construction, the regression results of the urban construction area are spatially concentrated in the southeast and gradually gather in the surrounding underdeveloped areas, indirectly indicating that the urbanization level of Xining, Chengdu, and other urban circles in the east is relatively high, the original road network coverage is high, the traffic development is slow, and the speed of urban-rural integration in the southeast is significantly greater than the speed of traffic advantage change. At the same time, the demand for land transportation construction is gradually increasing in remote areas.

From the perspective of population density in the social development dimension (Figure 10e,f), the spatial distribution of the regression coefficient in 2015 and 2019 shows that the development of traffic infrastructure in saturated areas has slowed down, and the development focus has been on the western and southeast parts of the Qinghai-Tibet Plateau with underdeveloped traffic. Eastern Xining, Lanzhou, Chengdu, and other high-development areas with densely populated cities and towns, have a roughly formed traffic infrastructure construction scale. Akto County, Wuqia County, and other central towns in the northwest, where the population is relatively dense, bear the important function of connecting the county. With the increase in local population carrying capacity and aggregation ability, the change rate of traffic dominance in the western and southeast regions is faster than that of population growth, reflecting the different emphasis on traffic development in the region and the improvement of traffic inconvenience in remote areas.

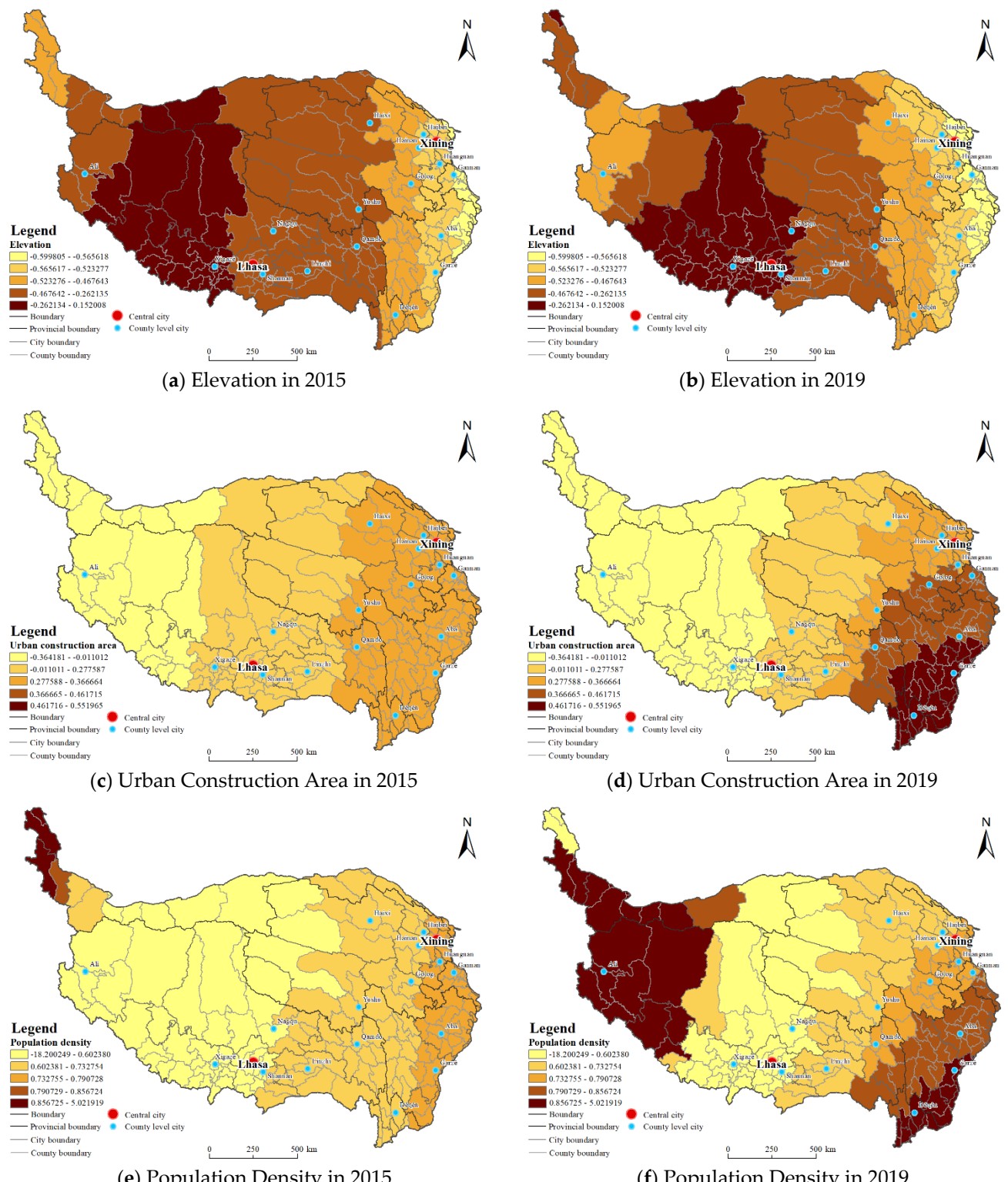

(**a**) Elevation in 2015      (**b**) Elevation in 2019

(**c**) Urban Construction Area in 2015      (**d**) Urban Construction Area in 2019

(**e**) Population Density in 2015      (**f**) Population Density in 2019

**Figure 10.** The spatial differentiation of the influencing factors of traffic dominance in the Qinghai-Tibet Plateau in 2015 and 2019. (**a**) distribution map of elevation regression results in 2015; (**b**) distribution map of elevation regression results in 2019; (**c**) distribution map of urban construction area regression results in 2015; (**d**) distribution map of urban construction area regression results in 2019; (**e**) distribution map of population density regression results in 2015; (**f**) distribution map of population density regression results in 2019.

### 3.2.2. Analysis of Influencing Mechanisms

With the promotion of the Belt and Road policy and infrastructure construction, and further promoting the economic and social development of the region and the improvement of people's living standards, the expansion process of the Qinghai-Tibet Plateau transportation network entered a period of rapid development from 2015 to 2019. Among them, elevation, urban construction area, and population density were all significant indigenous factors that affected the development of the traffic network and affect the traffic dominance of the Qinghai-Tibet Plateau in different ways.

Elevation is an important factor affecting the construction of transportation infrastructure. In areas with relatively flat terrain, accessibility is usually strong. In the highland areas of the central Qinghai-Tibet Plateau, such as the Qiangtang Plateau, traffic construction is difficult, time-consuming, and expensive, which is not conducive to the construction of infrastructures such as traffic roads, airports, and railway stations. Speeding up the construction of transportation infrastructure plays a vital role in strengthening the economic development and cultural exchanges between regions of the Qinghai-Tibet Plateau. Additionally, the regions with high altitudes in the Qinghai-Tibet Plateau are mostly mountainous areas, and the climate environment is harsh and changeable. If soil and water conservation is not paid attention to with the construction of transportation infrastructure, it is easy to damage the local ecological environment.

The urban construction area is an important embodiment of urbanization. Areas with less urban construction area are characterized by less supporting municipal infrastructure, relatively backward regional development, and lower urban vitality. The construction carrying capacity of large airports and railway stations is limited, which further affects the difference in traffic radiation effect, leading to the difference in traffic connection ability between counties, and ultimately reflects the difference in the spatial distribution of traffic dominance in the Qinghai-Tibet Plateau.

The greater the population density, the higher the degree of population aggregation; the stronger the carrying capacity of the population, the higher the travel demand for the region, the more labor available for the locals, and the greater the contribution to the construction of transport infrastructure. At the same time, the increase in population carrying capacity can strengthen the level of local transportation infrastructure and road network and can promote attraction to the region to increase population aggregation, creating a positive effect. Therefore, population aggregation will inevitably promote the development of a regional traffic network which can improve the traffic advantage of the region and its surrounding areas to a certain extent.

## 4. Discussion

The Qinghai-Tibet Plateau is a key development target of the western development strategy and the Belt and Road policy. In recent years, China has balanced the relationship between ecological environment protection, local cultural heritage, and economic development of the Qinghai-Tibet Plateau through ecological economic construction [55]. At present, the high-quality and sustainable development of the Qinghai-Tibet Plateau is going through an accelerated stage [56], but the uncoordinated economic development and unbalanced construction of transportation infrastructure in most regions have hindered the high-quality social and economic development of the Qinghai-Tibet Plateau and in-depth cultural exchanges with foreign countries. Therefore, it is even more important to explore the spatial distribution pattern and impact mechanism of Qinghai-Tibet Plateau traffic dominance, scientifically and rationally plan regional transportation development, and avoid the imbalance of transportation construction layout and the resulting ecological and environmental problems and siphon effects, etc. High-quality sustainable development has important theoretical significance and practical value. At the same time, as an area with an extremely fragile ecological environment, the overall transportation system in the Qinghai-Tibet Plateau needs to be planned and developed in a more coordinated, integrated, and sustainable way. Firstly, in the process of the rapid construction of trans-

portation infrastructure, it is necessary to strengthen the sustainable development concept of environmental and cultural protection. For example, the massive laying of basic roads may lead to an increase in carbon emissions and the blockage of ecological corridors, which will lead to problems such as air pollution [57] and impacts on the living environment of wild animals [10,13]. Therefore, promoting the green and sustainable development method guaranteed by the construction of transportation infrastructure [58] and balancing the transportation network model with the overall development goals of protecting the local ecology and inheriting traditional culture in the future are still the keys to realizing the green and sustainable development of the Qinghai-Tibet Plateau; aspects cannot be ignored in the development of sustainable urbanization [59]. Secondly, the integrated development strategy of comprehensive transportation should be implemented according to the development level of different regions. For areas with high levels of transportation advantages, such as Xining and Lhasa central city clusters, on the basis of maintaining a higher ecological awareness, a transportation network with high efficiency, environmental protection, and convenient transportation should be advocated [60], as well as trunk railways. The integrated construction of inter-city railways promotes the effective connection of internal and external traffic within the city and contributes to the construction of a modern comprehensive transportation system on the Qinghai-Tibet Plateau. For the central and southeastern regions, such as Golmud City, Xigaze City, and Deqen County, the transportation advantage transition areas relying on local energy, tourism, and other related industries, improve the combination of railway passenger and freight functions and promote the integration of transportation and manufacturing, circulation links, and tourism resources, strengthening the connection between transportation trunk lines and important industrial parks and tourist attractions, and form a traffic-driven pattern in which industries promote the development of transportation. For areas in remote western regions with relatively backward economic development, such as the Ali area and Akto County, where the level of transportation advantage is relatively low, the scope, scale, and characteristics of human activities should be combined to adapt to local conditions. The local expressway network should be planned, strengthening the connection and coordination with national roads, rural roads, and other modes of transportation so as to improve the level of equalization of urban and rural transportation public services and improve the travel conditions of residents in remote areas.

It is complicated to systematically evaluate the development quality and level of regional transportation; the index system and calculation results of the transportation superiority degree constructed in this paper mainly reflect the objective and actual situation of the construction and development of regional transportation infrastructure. In the follow-up research, when evaluating the development level of regional transportation, more factors such as resource industry, carbon emissions, and climate should be combined, and then from a sustainable perspective assessment, a more comprehensive, scientific, and reasonable network model should be planned and constructed. The research on the relationship between transportation advantages and multi-level factors was carried out in order to more accurately evaluate and discuss the regional comprehensive transportation advantages and provide a reference for the optimization of the transportation layout and sustainable development of the Qinghai-Tibet Plateau.

## 5. Conclusions

Based on the traffic network, stations, and relevant social and economic data, this paper discusses the spatial-temporal pattern evolution and influencing mechanism of traffic dominance in the Qinghai-Tibet Plateau by methods of analysis of the OD cost matrix network and geographically weighted regression, and the conclusions of this paper mainly include:

1. The three dimensions of road density, the influence of trunk lines, and spatial accessibility are used to comprehensively evaluate the level of traffic development on the Qinghai-Tibet Plateau. According to the evaluation result, at the county scale, there is

spatial heterogeneity and dependence in the traffic dominance degree of the Qinghai-Tibet Plateau. It is mainly manifested by an irregular spatial pattern decreasing from the central region of Xining to Lhasa as the axis, with the traffic superiority degree radially distributed along the axis. Compared with 2015, the number of counties with low levels of traffic development decreased from 49 to 45 in 2019, the number of counties with a relatively high level increased from 40 to 44, and the number of counties with a high level increased from 39 to 44, indicating that the effects of transportation infrastructure construction on the Qinghai-Tibet Plateau in the two periods were remarkable.

2. Elevation, urban construction area, and population density are the significant factors for the spatial difference of traffic dominance degree at the county scale in the Qinghai-Tibet Plateau. Considering the multiple impacts of the natural environment, land resources, and social development on traffic dominance, OLS regression analysis was carried out based on the principle of the least squares regression method, and elevation, urban construction area, and population density were selected as the three factors that have a linear relationship with traffic dominance. The geographically weighted regression (GWR) model was used to estimate the parameters of the three dominant factors of elevation, urban construction area, and population density in 2015 and 2019. The explanatory power of the three indicators on the traffic dominance of the Qinghai-Tibet Plateau increased year by year.

3. The effects of rapid urbanization and population concentration on the spatial and temporal differentiation of traffic dominance gradually increase, while the effect of elevation tends to weaken from 2015 to 2019. The influence of the three factors on the traffic dominance of the Qinghai-Tibet Plateau is different in time and space, and there are different spatial distribution patterns. From the elevation of the natural environment dimension, the negative impact of elevation on traffic dominance in the Qinghai-Tibet Plateau is weakening; from the perspective of urban construction area in the dimension of land resources, the traffic development in the Qinghai-Tibet Plateau is still dominated by the developed eastern urban construction areas, and the focus of road infrastructure is gradually expanding to the west; from the perspective of population density in the social development dimension, the development of traffic infrastructure in saturated areas has slowed down, and the development focus has been on the western and southeast parts of the Qinghai-Tibet Plateau with underdeveloped traffic.

**Author Contributions:** Conceptualization, D.W. and Z.W.; methodology, Z.W.; software, K.W. and H.F.; validation, H.C., H.W. and H.L.; formal analysis, K.W. and Z.W.; investigation, J.G. and J.X.; resources, D.W. and Z.W.; data curation, H.F. and K.W.; writing—original draft preparation, K.W.; writing—review and editing, D.W. and Z.W.; visualization, K.W. and H.F.; supervision, Z.W.; project administration, D.W. and Z.W.; funding acquisition, D.W. All authors have read and agreed to the published version of the manuscript.

**Funding:** This research was funded by the Second Tibetan Plateau of Scientific Expedition and Research Program (STEP), grant number 2019QZKK0608.

**Institutional Review Board Statement:** Not applicable.

**Informed Consent Statement:** Not applicable.

**Data Availability Statement:** The data presented in this study are available from the corresponding author (Z.W.) upon reasonable request.

**Acknowledgments:** The authors thank the Institute of Tibetan Plateau Research and the Institute of Geographic Sciences and Natural Resources Research, Chinese Academy of Sciences, for providing basic geographic and thematic data on the Tibetan Plateau. The authors would also like to thank the editors and anonymous reviewers for their thorough and valuable comments and suggestions.

**Conflicts of Interest:** The authors declare no conflict of interest.

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
