# Peer review of "Spatial-Temporal Evolution and Influencing Mechanism of Traffic Dominance in Qinghai-Tibet Plateau"

_sustainability, doi:10.3390/su141711031_

Round 1

Reviewer 1 Report

It's an interesting article about transportation in Qinghai_Tibet Plateau region. Overall the findings are reasonable and convincing. The methods proposed in the article are also suitable for this type of research. A few comments:

1. Socio-economic indicators in table 1 are coming from two books in year 2016 and 2020, but the study period is year 2015 and 2019. Would there be any adjustment to that? Why not using socio-economic indicators of year 2015 and year 2019?

2. The third finding in the abstract is not very clear. What is population carrying capacity enhancement?

3. This paper studies elevation as one negative indicator for traffic dominance. But for positive indicators, the paper used six indicators. There is a huge difference between the two. Would this affect the credibility of the conclusion? To be more specific, there are many other negative indicators, for example, terrain, weather, accessibility to water/electricity etc.

4. I think the authors need to clearly explain what is “Lh” in table 2.

Author Response

请参阅附件。

Reviewer 2 Report

The work seems to me basically OK. No comments to the methods or the results. However there is one question I would like to see in the paper:

How does road construction influence the ecology and the society?

This is an area that has always been very remote, and the question comes up: Should the culture and nature of this area perhaps be left alone? Are there unique species and ecosystems here that are sensitive to modernization? Are there cultures that should be allowed to continue the same way they have done for centuries?

I don't know, and maybe there is no right  answer to this. But the questions should be mentioned.

Reviewer 3 Report

This particular topic has already been sufficiently covered by previous research. This work presents no progress in comparison to work already published.

It also contains significant flaws in designing transport infrastructure, in particular in relation to sustainable transport modes.

Round 2

Reviewer 3 Report

In Discussion authors recommended: expanding the density of freeways, national roads, provincial roads and county roads can effectively improve the capacity of transportation supply, transportation accessibility and external connection in Qinghai-Tibet Plateau.

Constructing more roads is not a sustainable solution. The analysis in this paper needs to be improved and suggestions should be in line with sustainability and environmental protection.

Intensified economic activities contribute to the lower environmental performance of inland transport. This negative influence which cannot be eliminated, but can only be mitigated by implementing environment-friendly policies. GHGs impact human health remarkably and inequitably because the minority of car owners generate pollution that also affects the less-privileged majority of Chinese society. To break this loop, the country needs a wise investment plan directed toward less environmentally harmful transport means such as rail. Expanding roads is not a sustainable solution.

I refer the authors to the paper: 

Sustainability assessment of inland transportation in China: A triple bottom line-based network DEA approach

Round 3

Reviewer 3 Report

The authors have responded to my concerns satisfactorily. I recommend acceptance.